# Design and Test of an Automatic Navigation Fruit-Picking Platform

Shaojiong Huang [1], Kaoxin Pan [1], Sibo Wang [1], Ying Zhu [1], Qing Zhang [1,*], Xin Su [2] and Hongjun Yu [2]

1   College of Engineering, China Agricultural University, Beijing 100083, China; shjhuang@21cn.com (S.H.)
2   Beijing Institute of Space Launch Technology, Beijing 100076, China
*   Correspondence: zhangqingbit@cau.edu.cn

**Abstract:** With the development in agricultural mechanization and information technology, orchard agricultural machinery is also constantly pursuing the goal of intelligence and efficiency. Fruit picking is the most labor-intensive part of the orchard harvesting process. In order to resolve the problems of high labor intensity, low picking efficiency, and labor shortage when harvesting dwarf high-density orchards, an automatic navigation fruit-picking platform with voice control was developed in the present study. First, the platform utilized a voice-controlled high-level extendable working platform and a fruit-box-lifting device to adapt to varying orchard planting row spacing and enable convenient fruit box loading and unloading. Second, an automatic navigation system, which employed China's Beidou navigation satellite system to acquire positional data and the Stanley algorithm for path-tracking control, was implemented. Third, the fruit-picking platform with automatic navigation system was fabricated and assembled, its outer wheel steering was measured to have a maximum angle of 30.3° and an average minimum turning radius of 4.5 m, meeting the turning radius requirements under orchard conditions. Finally, automatic navigation tests of the developed platform were performed in the orchard conditions. The results indicated that the platform could maintain a straight-line path with a maximum lateral deviation of 101.5 mm and a maximum absolute average deviation of 44.1 mm at 0.4 m/s. Under the U-shaped paths, the measured maximum lateral deviation was 148.6 mm and the maximum absolute average deviation was 57.2 mm. The navigation accuracy was sufficient to meet the requirements for the harvesting platform's routine operation in the orchards.

**Keywords:** picking platform; automatic navigation; voice control; path tracing

## 1. Introduction

China is a major producer of fruits and has been at the forefront of the world's fruit production in terms of yield and planting area since the 20th century [1]. In 2021, the total planted area with fruits in China reached 166.7492 million acres, and the yield reached 89.6154 million tons, both of which exceed half of the world's total production [2]. Fruit picking is one of the most labor-intensive processes in fruit production [3]. Currently, manual picking is still the main method of fruit picking in China, and is limited by high labor intensity, low picking efficiency and economic benefits, and significant safety hazards, which severely restrict the industrialization of fruit production [4,5].

Based on the current situation of fruit cultivation [6–9], it is necessary to design a fruit-picking platform that is highly mechanized and intelligent with good applicability to assist manual picking and improve picking efficiency [10–18]. The development of fruit-picking platforms in China started relatively late, and there is still a large gap in functionality, operational efficiency, and applicability compared with the advanced picking platforms [19–26]. Therefore, how to integrate existing intelligent technologies into harvesting platforms is a very practical issue.

Thomas Bak et al. applied automatic navigation technology to agricultural machinery and designed and manufactured an unmanned field test platform. The platform used four-wheel steering and had good maneuverability in the field. The field test results showed that the maximum tracking error was less than 1.6 cm when the platform was traveling at a speed of 0.2 m/s; at a speed of 1.6 m/s, the maximum tracking error was less than 10.7 cm [27]. Thomas Bell et al. from Stanford University used the CP-DGPS navigation positioning method to apply the automatic navigation system to the John Deere 7800 tractor. The navigation system mainly includes an automatic navigation control terminal, an automatic steering controller, a GPS satellite receiver, etc. Based on this, Thomas Bell et al. studied the tracking control algorithm for curved paths and the experimental results showed that the average tracking error was around 5 cm [28]. Many other researchers have also applied technologies such as automatic navigation and speech recognition to agricultural machinery [29–32]. Furthermore, most of the fruit-picking platforms in China are manually operated. Yang et al. designed a multifunctional, fully hydraulic orchard operating platform. The platform has a scissor-type lifting structure that can adapt to the picking of fruits at different heights of fruit trees and also adopts a stretchable workbench that can adapt to the picking of fruits in a large range of row spacing [33]. Hou Zhiwei et al. developed the 3GP-80 multi-functional wheeled work platform. The platform adopts a wheeled walking mode and the driving mechanism and lifting mechanism are completed through a hydraulic control system. The hydraulic motor is used to achieve stepless speed regulation and the maximum vertical lifting height can reach 1.5 m. The expansion device is used to increase the lateral area of the work platform [34]. However, those platforms lack intelligence and require manual operation of the platforms to get to the relevant work area. For fruit orchards with larger planting distances, the fruit-picking process is more cumbersome. For instance, when the fruit box is full, the operators have to stop picking and move the fruit box to the ground before continuing the picking work, resulting in low efficiency and high labor intensity.

In order to reduce the labor intensity and enhance the intelligence of orchard picking machinery, the present study creates an automatic navigation fruit-picking platform by combining the advantages of manually operated picking platforms. The platform can be navigated autonomously in the orchard without manual operation. Furthermore, with a voice control system, the extension of the extendable work platform and the lifting of the fruit-box-lifting device can be conducted automatically, bringing the benefits of improving work efficiency and reducing labor intensity.

## 2. Materials and Methods

### 2.1. Machine Structure and Working Principle

2.1.1. Machine Structure

The fruit-picking platform mainly consists of six parts, i.e., the driving system, steering system, braking system, chassis, hydraulic system, and working platform. The overall structure of the platform is shown in Figure 1. The driving system consists of a battery, a drive motor, a differential, a rear axle transmission, and wheels. The steering system adopts full hydraulic steering containing a hydraulic steering system and an electric steering wheel. The steering cylinder and hydraulic steering gear are connected by oil pipes.

The chassis serves as the supporting structure, with a high-level extendable working platform mounted on the top and fruit-box-lifting devices installed at the front and rear ends. A low-level working platform, driving system, and electrical control cabinet are located at the bottom. The hydraulic system includes two parts, i.e., the steering hydraulic system and the working hydraulic system. The working hydraulic system provides power to the fruit-box-lifting device and the high-level extendable working platform, while the steering hydraulic system provides hydraulic assistance when the platform is turning.

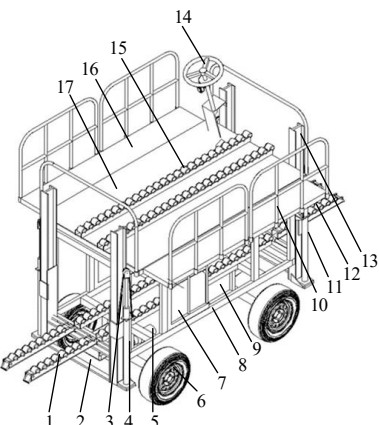

**Figure 1.** Structural diagram of automatic navigation fruit-picking platform. 1, 12 and 15 represent the fruit box transportation track, 2 represents the support frame, 3 represents the chains, 4 and 11 represent the hydraulic cylinder, 5 represents the drive motor, 6 represents the wheel, 7 represents the low-level working platform, 8 represents the body support, 9 represents the electric control cabinet, 10 represents the guard rail, 13 represents the slide rail, 14 represents the electric steering wheel, 16 represents the expansion pedal, and 17 represents the high-level working platform.

The high-level extendable working platform consists of left and right expansion pedals, expansion hydraulic cylinders, sliding rails, and a workbench. The expansion pedals are driven by hydraulic cylinders to extend left and right, as shown in Figure 2. The low-level working platform consists of a seat, a backrest, and hinges, which are fixed to the chassis by hinges for workers to pick up low-level fruits.

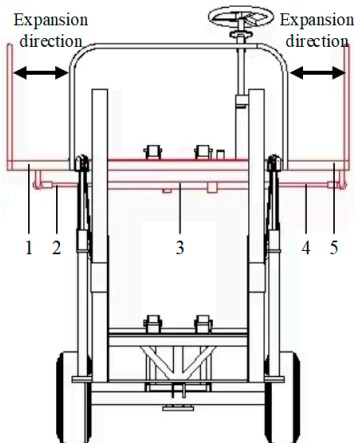

**Figure 2.** Structural diagram of the high-level expendable working platform. 1 and 5 represent the expansion pedals, 2 and 4 represent the hydraulic cylinders, and 3 represents the sliding rails.

The box-lifting device consists of a support frame, a sliding frame, a fruit box transport device, and a lifting hydraulic cylinder. There is one fruit-box-lifting device installed at the front and the rear of the platform, respectively, making it easy to load and unload the fruit boxes. The fruit box transportation route is shown in Figure 3. In order to make the picking platform more intelligent, automatic navigation technology and voice control technology are used based on the overall design. The picking platform can achieve autonomous driving in the orchard through the automatic navigation system, the fruit-box-lifting device can be lifted, and the high-level extendable working platform can be extended through voice control. The electrical control cabinet is used to place electronic components, receive external signals, and control the stable operation of each system.

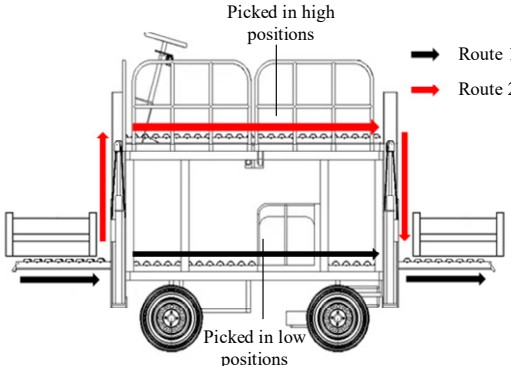

**Figure 3.** Fruit box transportation route.

2.1.2. Working Principle

The fruit-picking platform is driven by an electric motor and uses an automatic navigation system to achieve autonomous driving on a specified path. After reaching the predetermined position, the operator can control the platform's brake and the extension of the high-level extendable working platform through voice control, making it easier for manual fruit picking. When the fruit box is full, it can be unloaded by the fruit-box-lifting device.

The working principle of the manual steering system is shown in Figure 4. In order to ensure the safety and reliability redundancy of the platform and meet manual steering requirements in some scenarios, a manual steering system is also designed for the fruit-picking platform. When manual steering is required, the operator can control steering wheel 2 to drive the steering column, which drives the follow-up valve inside the full hydraulic steering gear 3 to rotate. Under the action of hydraulic pump 1, hydraulic oil flows into the steering hydraulic cylinder 7 through the follow-up valve to push the piston rod to extend and contract, thereby driving the steering tie rod to realize the steering of the picking platform. When the driver stops turning the steering wheel, the hydraulic oil flows through the follow-up valve of the steering gear and directly back to oil tank 10, and the steering hydraulic cylinder remains in its original position. During path tracking, when the picking platform reaches the turning area, the Beidou navigation system sends the obtained position information to the industrial control computer. After receiving the information, the industrial control computer sends the corresponding control program to the microcontroller. After the microcontroller runs the program, it sends a specific command to drive the steering motor to complete the steering.

The working principle of the extension platform and fruit-box-lifting hydraulic system is shown in Figure 5. Hydraulic pump 1 starts to supply oil to the hydraulic system and unloading valve 2 controls the system pressure. The solenoid directional valves 3, 4, 5, and 6 are in the middle position. When the front lifting command is received, the solenoid directional valve 3 switches to the right position and the piston rod of the front fruit-box-lifting hydraulic cylinders 7 and 8 extends to drive the fruit-box-lifting device to rise. When the stop front lifting command is received, the solenoid directional valve 3 returns to the middle position, hydraulic cylinders 7 and 8 stop moving, and the fruit-box-lifting device stays in place. When the front lowering command is received, the solenoid directional valve 3 switches to the left position and the piston rod of the front fruit-box-lifting hydraulic cylinders 7 and 8 retracts to drive the fruit-box-lifting device to descend. The working principles of the remaining hydraulic cylinders 9, 10, 11, and 12 are the same, respectively realizing the extension of the platform and the lifting of the rear fruit-box-lifting device.

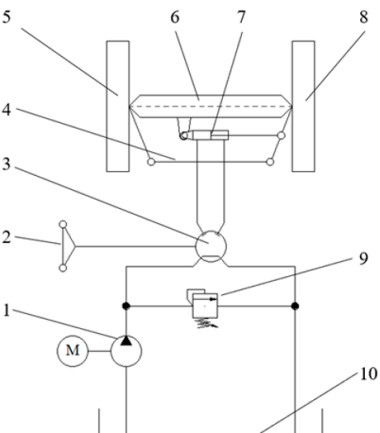

**Figure 4.** Schematic diagram of the steering system. 1 represents the hydraulic pump, 2 represents the electric steering wheel, 3 represents the full hydraulic steering gear, 4 represents the steering tie rod, 5 represents the left steering wheel, 6 represents the steering front axle, 7 represents the steering hydraulic cylinder, 8 represents the right steering wheel, 9 represents the unloading valve, and 10 represents the oil tank.

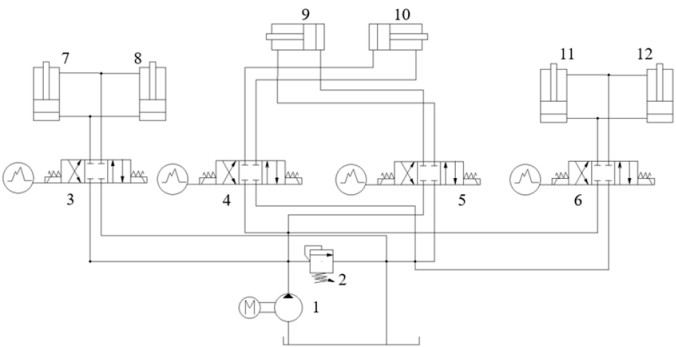

**Figure 5.** Schematic diagram of the hydraulic system of the extendable working platform and fruit box. 1 represents the hydraulic pump; 2 represents the unloading valve; 3, 4, 5, and 6 represent the solenoid directional valve; 7, 8, 11, and 12 represent the lifting hydraulic cylinders; 9 and 10 represent the extension hydraulic cylinders.

*2.2. Key Component Design*

2.2.1. Design of High-Level Extendable Working Platform

In order to facilitate operators to pick fruits at different heights and to meet the needs of multiple operators working at the same time, the platform has a double-sided, double-height, four-workstation design consisting of a high-level working platform and a low-level working platform. Since most fruits grow at the top of trees, the width of the high-level working platform is crucially important. A narrow high-level working platform would limit the working range of the operators, and frequent adjustments of the distance between the picking platform and the trees would greatly reduce work efficiency. To improve the adaptability and efficiency of the picking platform, a high-level working platform with an adjustable width is designed. To make the structure of the picking platform more compact, the high-level working platform is equipped with an expansion pedal. When the work width is sufficient, the expansion pedal retracts below the working platform and when the work width is not enough, operators can control the left and right expansion pedals to extend them. As shown in Figure 6, the left and right expansion pedals 1 and 5 are installed on slide rail 3 and the middle of the workbench 4 is equipped with a pulley. Operators control the telescoping motion of the telescopic hydraulic cylinders 2 and 6 to make the left and right expansion pedals 1 and 5 extend and retract along slide rail 3.

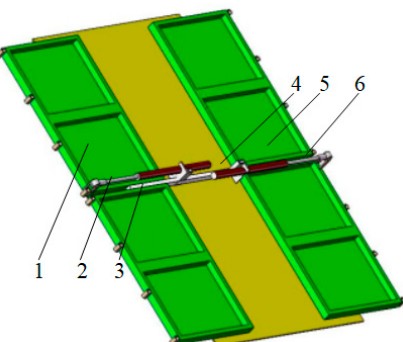

**Figure 6.** Structural diagram of the high-level extendable working platform. 1 and 5 represent expansion pedals, 2 and 6 represent the hydraulic cylinders, 3 represents the slide rail, and 4 represents the workbench.

To ensure the safety of the operator during the stretching and shrinking movement of the left and right extension pedals, the operator needs to wait for the extension pedals to fully extend before standing on them to carry out picking work. Based on the load of the working platform stretching hydraulic cylinder and the actual requirements of the orchard operation, the specific parameters of the selected single-rod double-acting hydraulic cylinder are shown in Table 1.

**Table 1.** Hydraulic cylinder parameters of the high-level extendable working platform.

| Parameters | Value |
| --- | --- |
| Hydraulic cylinder diameter (mm) | 50 |
| Piston rod diameter (mm) | 25 |
| Maximum working pressure (MPa) | 20 |
| Piston rod speed (m/s) | 0~0.6 |
| Piston rod stroke (mm) | 380 |

2.2.2. Design of the Fruit-Box-Lifting Devices

To facilitate the loading and unloading of fruit boxes on the high-level working platform and reduce labor intensity for operators, front and rear fruit-box-lifting devices were designed as shown in Figure 7 and installed at the front and rear ends of the picking platform, as shown in Figure 8. Before picking, the workers lift the empty fruit boxes to the high-level working platform using the fruit-box-lifting device, and after the fruit boxes are filled, they are unloaded with the help of the fruit-box-lifting device. This lifting structure does not affect the normal operation of the picking platform and can effectively improve work efficiency. The front and rear fruit-box-lifting devices use a guide rail lifting method, with support frame 1 installed at the front and rear ends of the picking platform and sliding frame 3 installed on support frame 1. The power generated by the extension and contraction of lifting hydraulic cylinder 5 drives chain 2 to pull the sliding frame up and down on the support frame. The fruit box transport device 4 is hinged on the sliding frame 3 and is equipped with pulleys for easy loading and unloading of the fruit boxes.

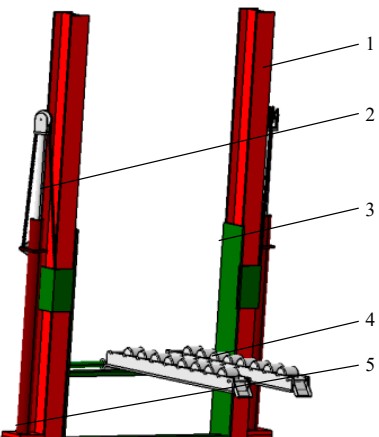

**Figure 7.** Structural diagram of the fruit-box-lifting device. 1 represents the support frame, 2 represents the chains, 3 represents the sliding frame, 4 represents the fruit box transport device, and 5 represents the lifting hydraulic cylinder.

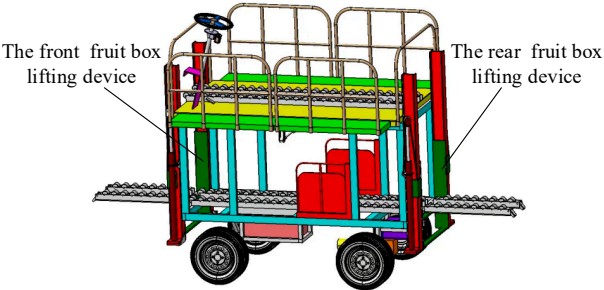

**Figure 8.** Installation diagram of the fruit-box-lifting device.

### 2.2.3. Design of the Drive System

The fruit-picking platform is driven by an electric power system that uses a battery to supply power to the drive motor. The drive system consists of a transmission system, a drive motor, and a battery. The power transmission is shown in Figure 9. The battery supplies power to the drive motor and electric motor. The drive motor provides power for the platform to travel, while the electric motor drives the gear pump to provide power for the hydraulic system. The differential changes the direction of power transmission and increases torque through deceleration.

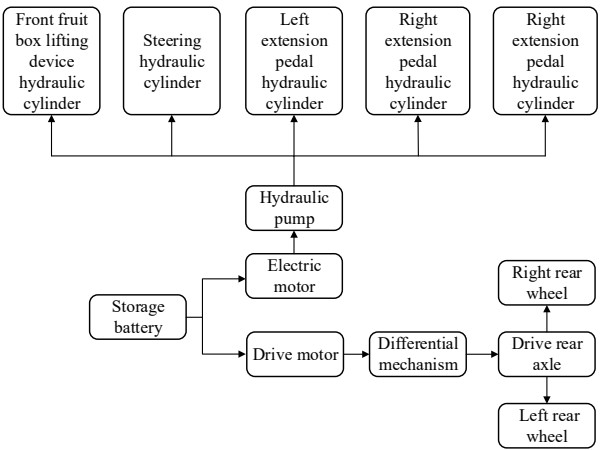

**Figure 9.** Diagram of power transmission.

When the operator drives the fruit-picking platform during picking operations, the driving speed needs to be set in combination with the operator's picking speed to ensure a high picking rate. At the same time, when reaching an area with dense fruit growth, the operator needs to apply the brake to the picking platform through the voice control system before picking. The voice recognition process takes about 1.6 s and the distance traveled by the picking platform at this time should not exceed the operator's picking range (i.e., the length of the working platform). Therefore, the design of the picking platform's driving speed can be calculated using Equation (1).

$$v \leq \frac{S}{t_1 + t_2 + t_s}, \tag{1}$$

where $v$ = the driving speed of the picking platform (m/s), $S$ = the operator's picking range (m), $t_1$ = the average picking time per working platform length (s), $t_2$ = the time consumed by voice recognition (s), and $t_s$ = the correction time (s).

The calculated driving speed is $v \leq 0.41$ m/s, and for convenience of subsequent calculations and relevant experiments, the design speed is set as 0.4 m/s.

The main forces the platform experiences during the picking operation include ground rolling resistance caused by its own weight and slope resistance caused by uneven road surfaces. The motor power is mainly used to overcome the rolling resistance, $F_f$, and the slope resistance, $F_p$. The application scenario of the picking platform is standardized orchards in plain areas of China. The road inside the orchard is flat, with a maximum slope of no more than 5°, which is calculated as 5° here.

The balance equation for the picking platform during operation is shown in Equation (2).

$$F_q = F_f + F_p = \mu m g \cos \theta + m g \sin \theta, \tag{2}$$

where $F_q$ = the driving force (N), $F_f$ = the ground frictional resistance (N), $F_p$ = the slope resistance (N), $m$ = the mass of the entire machine (kg), $g$ = the gravitational acceleration (m/s$^2$), $\mu$ = the rolling friction coefficient, and $\theta$ = the maximum slope (°).

Substituting the relevant parameters into Equation (2) and calculating gives a driving force, $F_p$, of approximately 2684.5 N.

The driving wheel speed can be calculated using Equation (3).

$$n = \frac{60v}{2\pi r}, \tag{3}$$

where $n$ = the driving wheel speed (rpm), $v$ = the picking platform traveling speed (m/s), and $r$ = the driving wheel radius (m).

Substituting the relevant parameters into Equation (3), the driving wheel speed is calculated to be approximately 11.75 rpm.

The DC motor power can be calculated using Equation (4).

$$p_q = F_q v. \tag{4}$$

Substituting relevant parameters, $p_q$ can be approximated as 1.074 kW. Based on the power requirements and usage of the picking platform, a DC series wound motor with a reduction ratio of 50 is selected and the forward and backward movements of the picking platform are controlled by the motor's forward and reverse controls. The main technical parameters of the selected DC series wound motor are shown in Table 2.

**Table 2.** Drive motor parameters.

| Parameters | Rated Voltage (V) | Rated Speed (r/min) | Output Speed (r/min) | Rated Power (kW) |
|---|---|---|---|---|
| valve | 60 | 3500 | 700 | 1.5 |

Based on the selected rated voltage of the drive motor, five 12 V lithium batteries are selected in series to power the drive motor. Considering the working time requirement of the picking platform, the platform needs to operate for at least 8 h after each charge. Therefore, the battery capacity, C, should be no less than 400 Ah, which is calculated as 50 A × 8 h. The main parameters of the selected batteries are shown in Table 3.

**Table 3.** Battery main parameter.

| Parameters | Rated Voltage (V) | Electric Capacity (Ah) | Operation Temperature (°C) | Overall Dimension |
|---|---|---|---|---|
| valve | 12 | 400 | −20~60 | 400 × 250 × 225 |

### 2.3. Design of the Voice Control System

2.3.1. Functional Requirements and Principles of the System

The voice control system consists of two parts, i.e., the operation voice control system and the walking voice control system. The operation voice control system includes the voice control of the front and rear fruit-box-lifting devices and the left and right extension pedals of the high-level working platform. During automatic navigation and picking operation, the walking voice control system can be used to brake the picking platform. After completing the picking operation, the picking platform can be controlled to continue moving forward. The structure of the voice control system is shown in Figure 10.

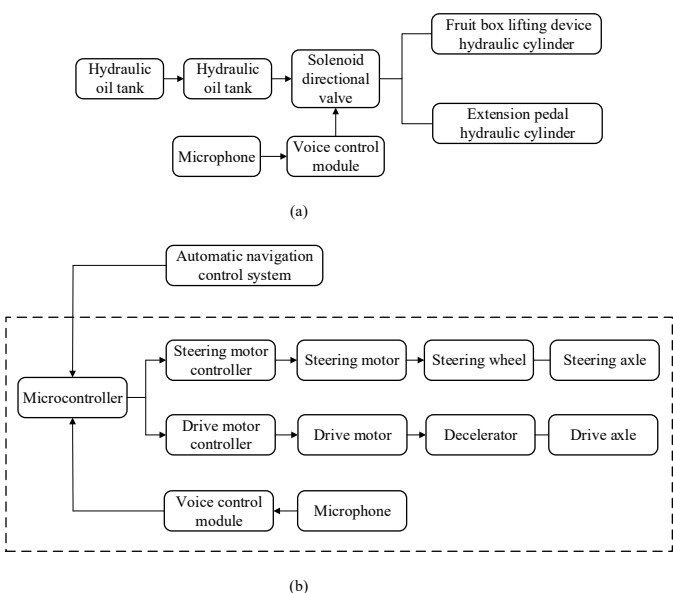

**Figure 10.** Structural composition of the voice control system. (**a**) Structural composition diagram of the operation voice control system. (**b**) Structural composition diagram of the walking voice control system.

To ensure the effectiveness of voice control, the system should accurately recognize the voice keywords of non-specific personnel with a recognition accuracy of no less than 85%. Moreover, the system should respond quickly, with a delay of no more than 2 s.

2.3.2. Design of the Voice Control System

(1)　Voice control chip

The voice control system utilizes the LD3322 as the main control chip for the voice recognition module. The LD3322 chip has a rich keyword list that can meet the required voice control commands. The chip comes with an onboard microphone that can convert the recognized sound signal into an electrical signal.

The LD3322 voice control module is controlled through the Intelligent Public Platform for Logic Editing. The GPIO pins are set to default low level, and the corresponding relay can be controlled by setting the GPIO pin to high or low. The picking platform operation system can control the corresponding solenoid directional valve by setting the GPIO pin. The UART1_TX pin can send hexadecimal serial data and the picking platform walking voice control system can send the corresponding hexadecimal command to the microcontroller through the UART1_TX pin.

The LD3322 voice control principle is shown in Figure 11. First, the keywords are edited on the above platform, and the platform generates the keywords in the form of pinyin letter strings. The generated voice control program is written into LD3322. When working, the voice keywords spoken by the operator are input through the MIC, and the LD3322 performs frequency spectrum analysis on the input keywords, extracts the sound features, and compares and matches them with the written keyword features. After finding the best match keyword, the matching result is sent to the MCU. At this point, the MCU converts the keyword into the corresponding instruction to control the solenoid valve and motor for the corresponding action.

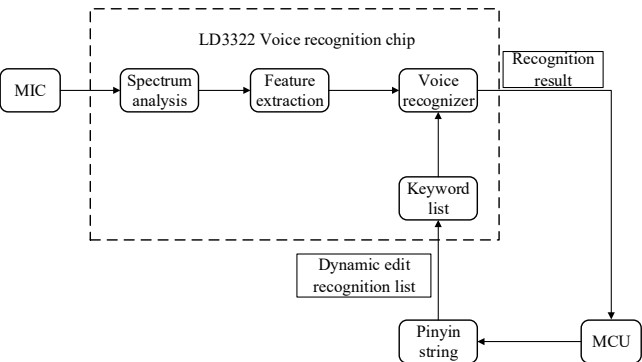

**Figure 11.** Schematic diagram of voice control.

(2)   Voice control system Debugging

Based on the working principle of the voice control system for the picking platform, the following steps can be taken to develop a list of voice recognition keywords and set the corresponding pin levels for voice recognition output. The high and low level pins correspond to the corresponding actions of the hydraulic cylinder. The list of voice recognition keywords, voice recognition output, and hydraulic cylinder actions for the picking platform are shown in Table 4.

**Table 4.** Operation voice system control keywords.

| Voice Recognition Keywords | Voice Recognition Output | Hydraulic Cylinder Actions |
|---|---|---|
| "cai3 zhai1 ping2 tai2" | "wo3 zai4" | Waiting for commands |
| "qian2 sheng1" | GPIO_B3 high level | The front lifting hydraulic cylinder extends |
| "ting2 zhi3 qian2 sheng1" | GPIO_B3 low level | The front lifting hydraulic cylinder stops extending |
| "qian2 jiang4" | GPIO_B2 high level | The front lifting hydraulic cylinder retracts |
| "ting2 zhi3 qian2 jiang4" | GPIO_ B2 low level | The front lifting hydraulic cylinder stops retracting |
| "hou4 sheng1" | GPIO_B6 high level | The rear lifting hydraulic cylinder extends |
| "ting2 zhi3 hou4 sheng1" | GPIO_B6 low level | The rear lifting hydraulic cylinder stops extending |
| "hou4 jiang4" | GPIO_B7 high level | The rear lifting hydraulic cylinder retracts |
| "ting2 zhi3 hou4 jiang4" | GPIO_B7 low level | The rear lifting hydraulic cylinder stops retracting |

**Table 4.** *Cont.*

| Voice Recognition Keywords | Voice Recognition Output | Hydraulic Cylinder Actions |
|---|---|---|
| "zuo3 shen1" | GPIO_A25 high level | The left telescopic hydraulic cylinder extends |
| "ting2 zhi3 zuo3 shen1" | GPIO_A25 low level | The left telescopic hydraulic cylinder stops extending |
| "zuo3 suo1" | GPIO_A26 high level | The left telescopic hydraulic cylinder retracts |
| "ting2 zhi3 zuo3 suo1" | GPIO_A26 low level | The left telescopic hydraulic cylinder stops retracting |
| "you4 shen1" | GPIO_A27 high level | The right telescopic hydraulic cylinder extends |
| "ting2 zhi3 you4 shen1" | GPIO_A27 low level | The right telescopic hydraulic cylinder stops extending |
| "you4 suo1" | GPIO_A28 high level | The right telescopic hydraulic cylinder retracts |
| "ting2 zhi3 you4 suo1" | GPIO_A28 low level | The right telescopic hydraulic cylinder stops retracting |

In order to test the success rate and effectiveness of the voice control system, the relevant hardware is connected and tested. The test model of the overlap is shown in Figure 12. We can use wake-up words "cai3 zhai1 ping2 tai2" to put the voice control into working mode first, and then conduct experimental verification with other voice keywords. If the voice system does not recognize the keywords within 5 s, it will enter standby mode and need to be awakened again.

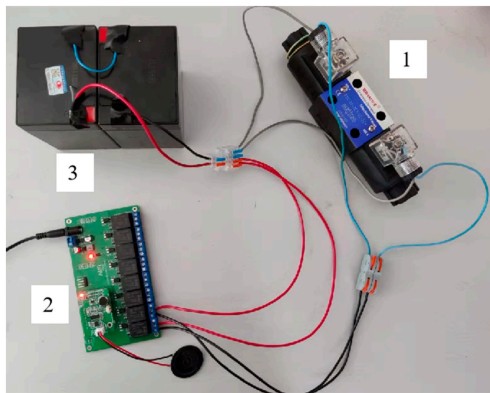

**Figure 12.** Voice test system of operation. 1 represents the solenoid directional valve, 2 represents the voice control module, and 3 represents the battery.

The experiments were conducted under two different background sound environments, namely, a quiet environment (35 dB in normal indoor conditions) and a noisy environment (when the machine is working normally in the orchard). Five representative groups of keywords were selected from Table 4 and tested 100 times each to determine the accuracy of the solenoid directional valve switching or disconnection recognition. The results are shown in Table 5. The average time required for accurate recognition was also recorded and are shown in Table 6.

**Table 5.** Recognition accuracy of the operation voice system in the two environments.

| Commands | Recognition Rate in the Quiet Environment (%) | Recognition Rate in the Noisy Environment (%) |
|---|---|---|
| "cai3 zhai1 ping2 tai2" | 93.0 | 90.0 |
| "qian2 sheng1" | 95.0 | 92.0 |
| "ting2 zhi3 qian2 sheng1" | 93.0 | 85.0 |
| "zuo3 suo1" | 96.0 | 85.0 |
| "ting2 zhi3 zuo3 suo1" | 90.0 | 83.0 |
| Average recognition rate | 93.4 | 87.0 |

**Table 6.** Recognition time of the operation voice system in the two environments.

| Commands | Recognition Time in the Quiet Environment (s) | Recognition Time in the Noisy Environment (s) |
|---|---|---|
| "cai3 zhai1 ping2 tai2" | 1.4 | 1.6 |
| "qian2 sheng1" | 1.0 | 1.3 |
| "ting2 zhi3 qian2 sheng1" | 1.6 | 1.8 |
| "zuo3 suo1" | 1.2 | 1.5 |
| "ting2 zhi3 zuo3 suo1" | 1.6 | 1.8 |
| Average recognition time | 1.4 | 1.6 |

Data in Table 5 show that the command recognition rate of the voice control system for the picking platform is relatively high in a quiet indoor environment, with an average recognition rate of 93.4%. However, in a noisy environment, the recognition rate is lower, with an average accuracy of 87.0%.

Data in Table 6 shows that the duration of voice recognition is associated with the length of the voice control keyword. The longer the recognition command, the longer the average recognition time. Longer recognition time is needed in a noisy environment compared with a quiet environment. For the same command, the recognition time in a noisy environment is 0.1–0.2 s longer than in a quiet environment.

Meanwhile, a walking voice recognition keyword list was compiled based on the working principle of the walking voice control system of the picking platform. The keyword list, voice recognition output, and corresponding walking actions for the picking platform walking voice control system are shown in Table 7.

**Table 7.** Walking voice system control keywords.

| Voice Recognition Keywords | Voice Recognition Output | Walking Actions |
|---|---|---|
| "cai3 zhai1 ping2 tai2" | "wo3 zai4" | Waiting for commands |
| "qian2 jin4" | 0 | Walk forward |
| "ting2 zhi3 qian2 jin4" | 1 | Stop walking forward |
| "hou4 tui4" | 2 | Walk backward |
| "ting2 zhi3 hou4 tui4" | 3 | Stop walking backward |

Connecting the relevant hardware and conducting experiments, the effectiveness of the voice control system was determined by judging the positive and reverse rotations of the motor. The experimental model is shown in Figure 13.

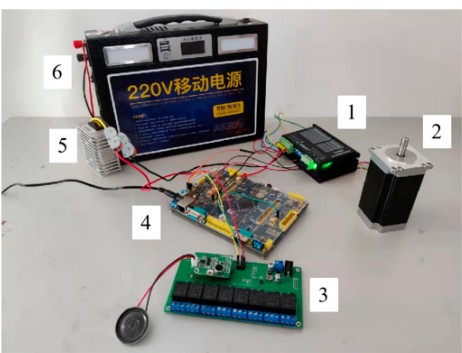

**Figure 13.** Voice test system for walking. 1 represents the motor driver, 2 represents the electric motor, 3 represents the voice control module, 4 represents the microcontroller, 5 represents the transformer, and 6 represents the battery.

In Table 7, each keyword was tested 100 times to verify the recognition results for the forward and reverse rotations of the steering motor. The results are shown in Table 8 and the average recognition time is recorded in Table 9.

**Table 8.** Recognition accuracy of the walking voice system in the two environments.

| Commands | Recognition Rate in the Quiet Environment (%) | Recognition Rate in the Noisy Environment (%) |
|---|---|---|
| "cai3 zhai1 ping2 tai2" | 96.0 | 92.0 |
| "qian2 jin4" | 95.0 | 89.0 |
| "ting2 zhi3 qian2 jin4" | 90.0 | 81.0 |
| "hou4 tui4" | 92.0 | 87.0 |
| "ting2 zhi3 hou4 tui4" | 97.0 | 90.0 |
| Average recognition rate | 94.0 | 87.8 |

**Table 9.** Recognition time of the walking voice system in the two environments.

| Commands | Recognition Time in the Quiet Environment (s) | Recognition Time in the Noisy Environment (s) |
|---|---|---|
| "cai3 zhai1 ping2 tai2" | 1.3 | 1.5 |
| "qian2 jin4" | 1.0 | 1.2 |
| "ting2 zhi3 qian2 jin4" | 1.4 | 1.7 |
| "hou4 tui4" | 1.1 | 1.3 |
| "ting2 zhi3 hou4 tui4" | 1.5 | 1.7 |
| Average recognition time | 1.2 | 1.5 |

Based on the data in Table 8, it can be concluded that the recognition rate of picking platform walking commands through voice control is relatively high in quiet indoor environments, with an average accuracy rate of 94.0%; while the recognition rate is lower in noisy environments, with an average accuracy rate of 87.8%.

Based on the data in Table 9, it can be concluded that the average recognition time of the walking voice control keywords is associated with the length of the keywords. The longer the recognition command, the longer the recognition time. It takes longer to recognize the keywords in noisy environments compared with quiet environments and, for the same command, the recognition time in noisy environments is 0.1–0.2 s longer than in quiet environments.

After analyzing the experimental results of the two voice control systems, it can be concluded that both systems can meet the requirements for normal operation of the picking platform.

### 2.4. Design of the Automatic Navigation System

2.4.1. Functional Requirements and Principles of the System

The automatic navigation system obtains the position information of the picking platform through high-precision satellite navigation equipment and plans the path based on the spacing between rows in the orchard and the distribution of fruit trees. Then, the path-tracking control algorithm is used to make the picking platform travel automatically along the planned path, thereby freeing the hands of operators and improving work efficiency. In order to ensure the stability of the picking platform during autonomous driving in the orchard, the automatic navigation system needs to obtain the position information of the picking platform in real time and accurately locate it. When it is detected that the picking platform has deviated from the planned path, the turning system can be controlled in real time to correct the deviation such that the picking platform always travels along the planned path.

The main components of the automatic navigation system include industrial computers, microcontrollers, monitors, Beidou navigation systems, angle sensors, and other components, as shown in Figure 14. Among them, the angle sensor needs to provide real-time feedback of steering angle information of the picking platform to the microcontroller, thus the angle sensor needs to meet requirements such as good shock resistance, high output accuracy, sturdiness, and durability.

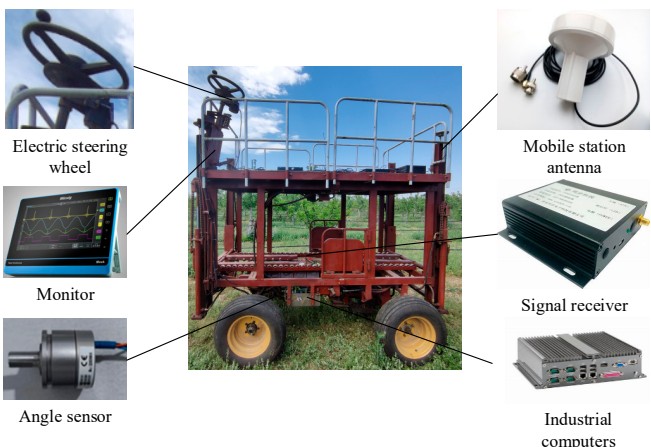

**Figure 14.** Components of the automatic navigation system.

### 2.4.2. Path-Tracking Control Algorithm

The automatic navigation fruit-picking platform adopts front-wheel steering and rear-wheel drive. Due to the low traveling speed under orchard operations, the kinematic constraints have a greater impact on the picking platform, while the dynamic characteristics have a smaller impact on the tracking control of the picking platform. Therefore, a kinematic model of the picking platform is established, taking the navigation coordinate system as the X-axis and Y-axis, and building an XOY rectangular coordinate system as shown in Figure 15. The kinematics model is called the bicycle model, and does not consider the movement of the vehicle in the vertical direction (Z axis direction). Meanwhile, it is assumed that the vehicle moves in a two-dimensional plane, and the left and right tires of the vehicle have the same steering angle and speed at any time, thus the motion of the left and right tires of the vehicle can be combined into one tire to describe.

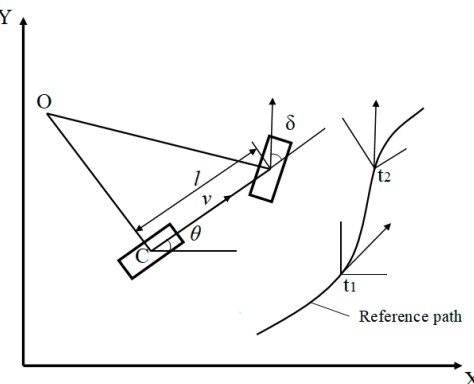

**Figure 15.** Schematic diagram of the kinematic model.

The kinematic Equation (5) of the picking platform can be obtained by deriving mathematical relationships between relevant parameters.

$$\begin{bmatrix} \dot{x}_c \\ \dot{y}_c \\ \dot{\theta} \end{bmatrix} = \begin{bmatrix} \cos\theta \\ \sin\theta \\ \tan\frac{\delta}{l} \end{bmatrix} v_c, \tag{5}$$

where $\dot{x}_c$ = the x-coordinate of point C, $\dot{y}_c$ = the y-coordinate of point C, $\theta$ = the heading angle in degrees with counterclockwise direction as positive (°), $\delta$ = the front wheel angle in degrees with counterclockwise direction as positive (°), $v_c$ = the speed of the picking platform (m/s), and $l$ = the wheelbase of the picking platform (m).

The Stanley algorithm is adopted as the path-tracking control algorithm using a kinematic model [35]. The core of the Stanley algorithm is to calculate the steering angle of the electric power steering system based on the deviation between the front wheel center and the desired path. By using the relative geometric relationship between the picking platform's positioning information and the desired path, the Stanley algorithm can obtain the control variable for the front wheel steering angle, which includes the steering angle, $\Psi_\sigma$, caused by the heading deviation, $\sigma$, and the steering angle, $\Psi_\theta$, caused by the lateral deviation, $e$. The principle is illustrated in Figure 16.

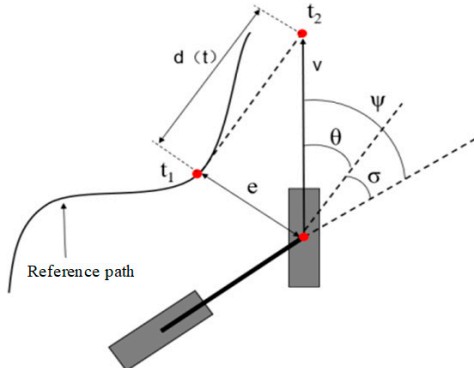

**Figure 16.** Schematic diagram of the Stanley algorithm.

Assuming that the lateral deviation is ignored, the steering angle of the front wheel is aligned with the tangent direction of the nearest point, $t_1$, on the reference trajectory, where $\sigma$ represents the angle between the harvesting platform and the tangent direction of point $t_1$. Therefore, we can derive Equation (6) as:

$$\Psi_\sigma(t) = \sigma(t), \tag{6}$$

where $\sigma$ = the heading deviation (°) and $\Psi_\sigma$ = the adjustment steering angle caused by the heading deviation $\sigma$ (°).

Assuming that the heading deviation is ignored, when the lateral deviation, $e$, increases, the required steering angle of the front wheel increases. Suppose the expected trajectory of the picking platform intersects with the tangent line of point $t_1$ on the reference trajectory at a distance of $d(t)$ in front of the front wheel, Equations (7) and (8) can be obtained based on the geometric relationship in Figure 16.

$$\Psi_\theta(t)k = \theta(t) = arctan\frac{e(t)}{d(t)} = arctan\frac{ke(t)}{v(t)}, \tag{7}$$

$$v = kd(t), \tag{8}$$

where $\theta(t)$ = the angle between the heading of the picking platform and the tangent direction at the nearest path point $t_1$ (°), $\Psi_\theta(t)$ = the adjusting angle caused by the lateral deviation $e$ (°), $e(t)$ = the lateral deviation of the picking platform at a given time (°), $d(t)$ = the distance between the intersection point $t_1$ and $t_2$ (m), $v(t)$ = the velocity of the picking platform (m/s), and $k$ = the gain parameter.

Taking both situations into consideration, Equation (9) is as follows:

$$\Psi(t) = \Psi_\sigma(t) + \Psi_\theta(t), \tag{9}$$

where $\Psi$ = the expected steering angle of the front wheel (°).

Additionally, based on the geometric relationship in the figure, the lateral deviation of the picking platform can be expressed as Equations (10) and (11).

$$\dot{e}(t) = -v(t)\sin \Psi_\theta(t), \tag{10}$$

$$\sin \Psi_\theta(t) = \frac{e(t)}{\sqrt{d(t)^2 + (e(t))^2}} = \frac{ke(t)}{\sqrt{v(t)^2 + (ke(t))^2}}, \tag{11}$$

where $\dot{e}(t)$ = the rate of change of lateral deviation.

### 2.4.3. The Working Process of the Automatic Navigation System

The automatic navigation system of the picking platform adopts a cross-row S-shaped walking path, as shown in Figure 17. The corresponding road width and fruit tree height are marked in the figure. When the picking platform performs autonomous navigation work, it transmits the posture information obtained by the Beidou navigation system to the industrial control computer, which sends corresponding instructions through the Stanley algorithm to control the turning, braking, and forward movements of the picking platform. During the turning process, the industrial computer sends corresponding instructions to the microcontroller that controls the steering hydraulic system, and then the microcontroller controls the steering hydraulic system for steering. At the same time, the angle sensor will provide real-time data feedback to achieve successful steering.

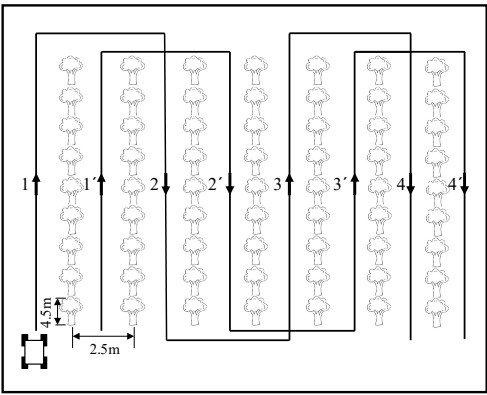

**Figure 17.** Automatic navigation driving.

## 3. Results and Discussion

### 3.1. Manufacturing and Testing the Fruit-Picking Platform

Manufacturing the Prototype

A prototype was developed by Xinnong Machinery Co., Ltd (Shijiazhuang, China). The overall dimensions of the autonomous navigation fruit-picking platform are 4435 mm × 1550 mm × 2960 mm, and the prototype is shown in Figure 18. The size and operating width of the picking platform can meet the basic requirements of operations in dwarf, highly dense orchards. The walking system of the fruit-picking platform is driven by a DC motor with a rated power of 1.5 kW. In order to verify the performance of the picking platform and promptly detect any problems that may occur during operation, relevant tests and verifications were conducted on the prototype.

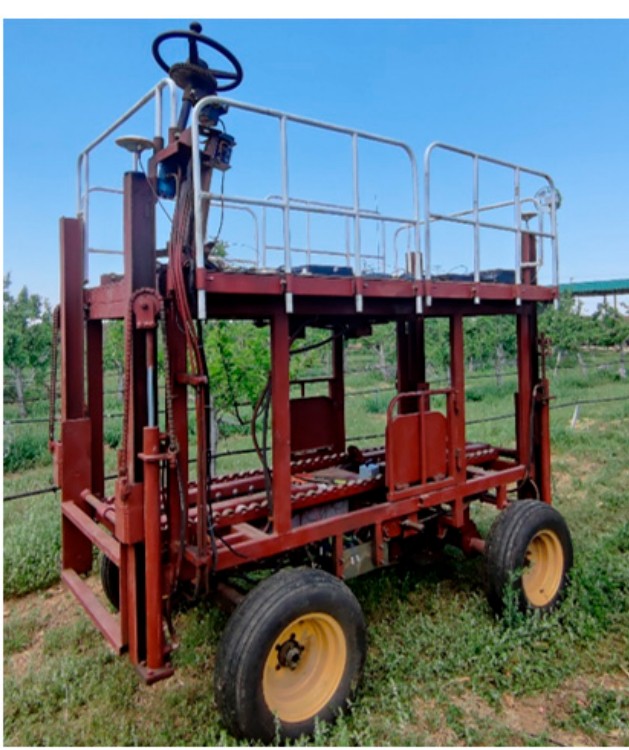

**Figure 18.** Photo of the fruit-picking platform.

*3.2. Testing the Prototype*

3.2.1. Test Conditions and Materials

The tests were conducted on the cement road and experimental orchard of Xinnong Machinery Co., Ltd., in Hebei Province. The weather was sunny with a south wind of level one and sunny with a southwest wind of level three. The experimental equipment and materials used in the performance tests included a leather tape measure, marking pen, digital universal protractor, stopwatch, etc.

3.2.2. Steering Function Tests of the Picking Platform

The purpose of the steering function tests of the picking platform is to verify whether the prototype's steering performance meets the design goals and requirements of cross-row operations in orchards. The tests include the steering angle test and the turning radius test.

(1)　Steering angle test

The picking platform is a front-wheel steering and rear-wheel drive system. Therefore, the steering test refers to relevant regulations for wheeled tractors and formulates a test plan according to the "JB/T7279-2008 Test Methods for Mechanical Steering Systems of Wheeled Tractors". Before turning the steering wheel, a marking pen was used to record the outer edge lines of the right front wheel and left front wheel of the picking platform. Then, the operator turned the steering wheel to the right and left turning dead angles in turn, recording the maximum steering angles of the left and right wheels and the steering wheel. The actual test scenario is shown in Figure 19.

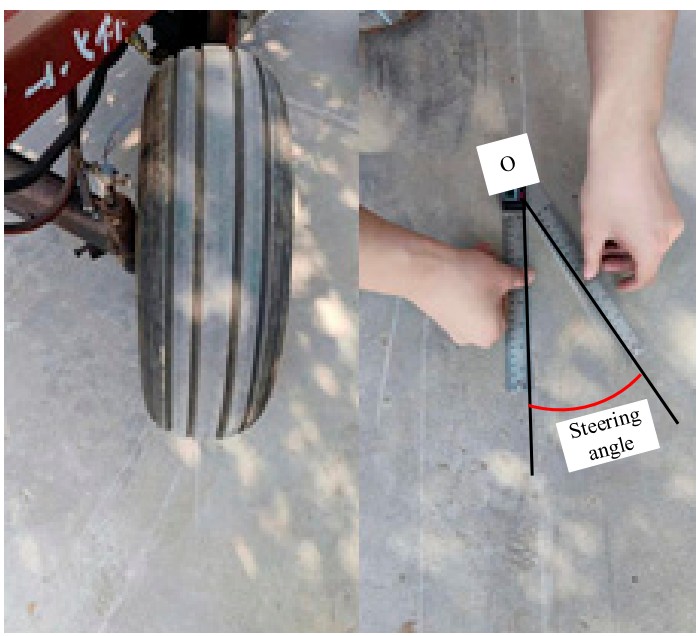

**Figure 19.** Steering angle test.

It is assumed that the clockwise rotation is positive and the counterclockwise rotation is negative. The recorded maximum steering angle data of the front wheels are shown in Table 10.

**Table 10.** Maximum steering angle test results.

| Parameter | Left Wheel Steering Angle (°) | Right Wheel Steering Angle (°) | Steering Wheel Angle (°) |
|---|---|---|---|
| Initial angle | 0 | 0 | 0 |
| Right steering maximum angle | 26.5 | 30.3 | 550 |
| Left steering maximum angle | −28.3 | −24.5 | −510 |

It can be seen that the maximum steering angles of the left and right turning wheels are 28.3° and 30.3°, respectively, which basically meets the requirement of the design value of 30°.

(2)　Turning radius test

The turning radius of the platform was tested according to GB/T 3871.5-2006 "Agricultural Tractor Test Code, Part 5: Turning Circle and Passing Circle". The turning circle radius obtained when the picking platform turns to the right was selected as the minimum turning radius data of the picking platform. Before testing, the driver kept the front wheels of the picking platform at the maximum right turning angle. When the picking platform turned a circle and left a turning circle impression, the diameter of the turning circle was measured and recorded using a leather tape measure. The test process is shown in Figure 20.

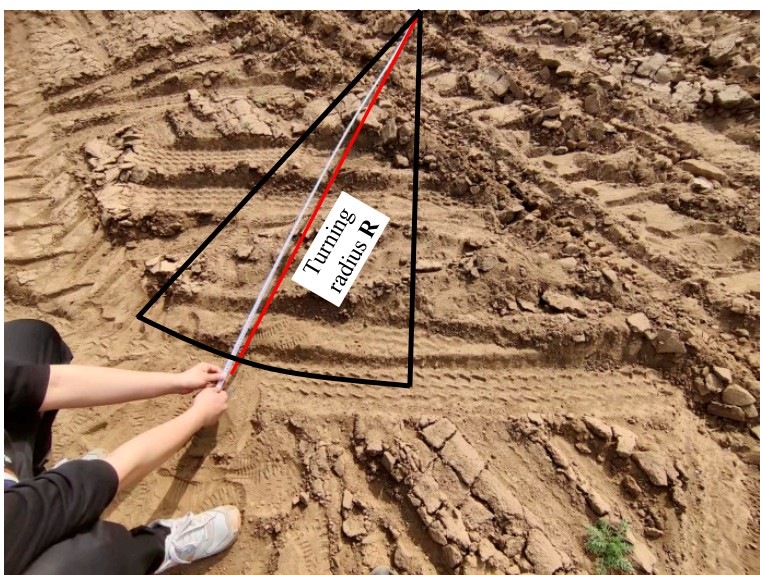

**Figure 20.** Turning radius test experiment.

Every test was conducted three times. The recorded turning circle diameters were processed to obtain the data in Table 11.

**Table 11.** Turning radius measurements.

| Number of Tests | Turning Radius of Outer Wheel (m) | Average Value (m) |
| :---: | :---: | :---: |
| 1 | 4.6 | |
| 2 | 4.3 | 4.5 |
| 3 | 4.7 | |

The average minimum turning radius of the outer wheels of the picking platform was 4.5 m. Based on the on-site measurement data of the test orchard, the maximum turning radius allowed in the orchard is 5 m. Therefore, the picking platform meets the design requirements for turning radius.

### 3.2.3. Voice Control Tests of the Picking Platform

The voice control module and the solenoid directional valve were installed on the prototype and the designed operation voice control system was tested in the orchard as shown in Figure 21.

After waking up the LD3322 main control chip with a wake-up word, the corresponding lifting commands were used to control the lifting devices of the front and rear fruit boxes of the picking platform to the maximum height. Each functional test was carried out five times. The results are shown in Table 12.

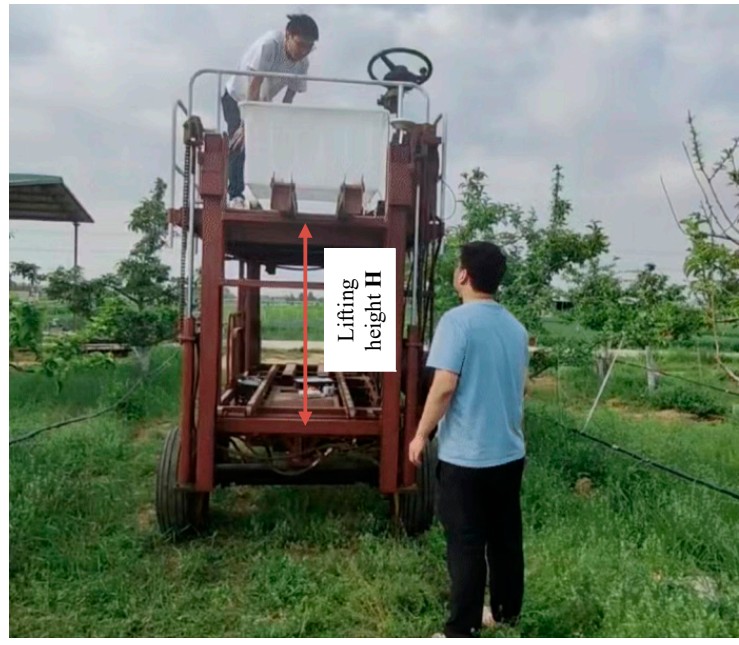

（a）

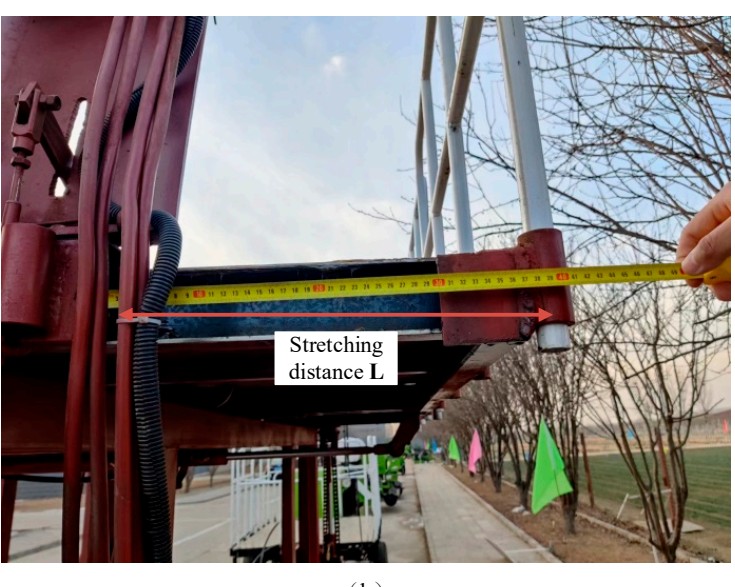

（b）

**Figure 21.** Voice control experiments. (**a**) Lifting test for the fruit-box-lifting device. (**b**) Extension test for the the high-level extendable working platform.

Table 12 shows that the absolute error of the maximum lifting height of the front and rear fruit box is less than 10 mm and the relative error is less than 1%, which can meet the requirements of orchard operations.

Using the corresponding stretching commands to control the left and right expansion pedals of the workbench to extend to the maximum distance, five sets of data were recorded and the results are shown in Table 13.

**Table 12.** Fruit-box-lifting device lifting height.

| Number of Tests | Theoretical Lifting Height (mm) | Lifting Height of Front Lifting Device (mm) | Absolute Error of Lifting Height (mm) | Relative Error of Lifting Height (%) | Lifting Height of Rear Lifting Device (mm) | Absolute Error of Lifting Height (mm) | Relative Error of Lifting Height (%) |
|---|---|---|---|---|---|---|---|
| 1 | 1940 | 1935 | 5 | 0.26 | 1934 | 6 | 0.31 |
| 2 | 1940 | 1937 | 3 | 0.15 | 1936 | 4 | 0.21 |
| 3 | 1940 | 1934 | 6 | 0.31 | 1935 | 5 | 0.26 |
| 4 | 1940 | 1936 | 4 | 0.21 | 1937 | 3 | 0.15 |
| 5 | 1940 | 1934 | 6 | 0.31 | 1938 | 2 | 0.10 |

**Table 13.** Pedal extension length.

| Number of Tests | Theoretical Extension Length (mm) | Left Extension Pedal Extension Length (mm) | Extension Length Absolute Error (mm) | Extension Length Relative Error (%) | Right Extension Pedal Extension Length (mm) | Extension Length Absolute Error (mm) | Extension Length Relative Error (%) |
|---|---|---|---|---|---|---|---|
| 1 | 380 | 376 | 4 | 1.05 | 378 | 2 | 0.52 |
| 2 | 380 | 375 | 5 | 1.32 | 376 | 4 | 1.05 |
| 3 | 380 | 373 | 7 | 1.84 | 375 | 5 | 1.32 |
| 4 | 380 | 374 | 6 | 1.57 | 377 | 3 | 0.79 |
| 5 | 380 | 375 | 5 | 1.32 | 376 | 4 | 1.05 |

Table 13 shows that the absolute errors of the length of the left and right expansion pedals of the workbench is within the range of 10 mm and the relative error is about 1%, which can meet the requirements of orchard operations.

### 3.2.4. Automatic Navigation Tests of the Picking Platform

To verify the effectiveness of the automatic navigation control of the picking platform, automatic navigation tests were conducted in the standardized demonstration orchards.

The paths were divided into two types, i.e., straight-line and U-shaped paths. The automatic navigation tests were carried out at a designed speed of 0.4 m/s on both paths. The total length of the straight-line path was set as 40 m, and the total length of the U-shaped path was 30 m. The test condition is shown in Figure 22.

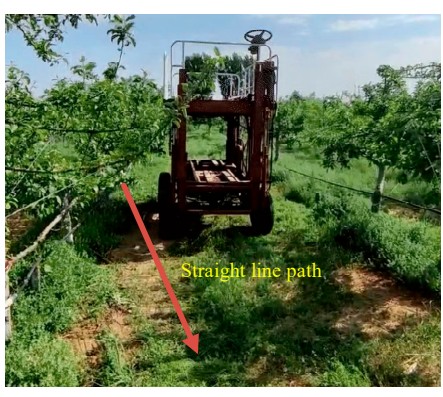

(a)

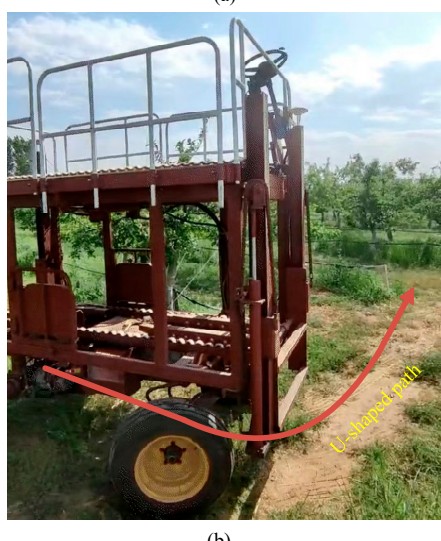

(b)

**Figure 22.** Automatic navigation tests. (**a**) Straight-line path navigation test. (**b**) U-shaped path navigation test.

The real-time coordinate information was recorded and processed against the preset path to calculate the actual lateral deviation. The deviation statistical results of the two paths are shown in Tables 14 and 15.

**Table 14.** Statistics of straight-line path lateral deviation.

| Number of Tests | Maximum Lateral Deviation (mm) | Average Lateral Deviation (mm) | Absolute Average Deviation (mm) | Standard Deviation (mm) |
|---|---|---|---|---|
| 1 | 100.3 | 3.1 | 39.0 | 20.5 |
| 2 | 93.2 | 1.4 | 39.5 | 17.1 |
| 3 | 101.5 | −1.4 | 44.1 | 19.5 |

**Table 15.** Statistics of U-shaped path lateral deviation.

| Number of Tests | Maximum Lateral Deviation (mm) | Average Lateral Deviation (mm) | Absolute Average Deviation (mm) | Standard Deviation (mm) |
|---|---|---|---|---|
| 1 | 129.2 | −6.9 | 51.9 | 23.4 |
| 2 | 140.4 | 1.5 | 57.2 | 28.0 |
| 3 | 148.6 | 9.7 | 57.0 | 26.4 |

Table 14 indicates that when the picking platform travels at a speed of 0.4 m/s, the maximum lateral deviation of the straight-line path is 101.5 mm and the maximum absolute average lateral deviation is 44.1 mm. Table 15 shows that when the picking platform travels at a speed of 0.4 m/s, the maximum lateral deviation of the U-shaped path is 148.6 mm and the maximum absolute average lateral deviation is 57.2 mm. The deviations are within a reasonable range and can meet the requirements of automatic navigation of the picking platform.

The data from the straight-line path-tracking test and the U-shaped path-tracking test were selected to draw the path-tracking effect diagram, as shown in Figure 23. The actual tracking path was good, indicating that the picking platform can meet the basic requirements of autonomous driving in the orchards.

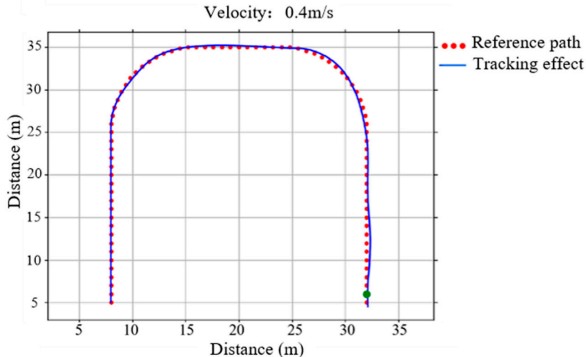

**Figure 23.** Effect of path tracking.

## 4. Summary

In order to resolve the problems of high labor intensity, low picking efficiency, and labor shortage when picking dwarf high-density orchards, an automatic navigation fruit-picking platform with a voice control system was developed in the study. The conclusions can be drawn as follows:

(1) A high-level extendable working platform and fruit-box-lifting device operated via voice control were adopted with the purpose of improving the applicability and work efficiency of the picking platform. The accuracy of the speech recognition and the response time of the voice control system were tested and verified via system testing.

(2) The assembly of the picking platform prototype was completed and the average minimum turning radius of the picking platform was 4.5 m, meeting the requirements

of the minimum turning radius in the orchard. Furthermore, the operating tests of the voice control system were conducted on the prototype. The results showed that both the maximum elevated height deviation of the front and rear fruit box and the maximum distance deviation of the high-level extendable working platform pedals were within 10 mm compared with the design value, meeting the requirements for fruit box loading and unloading and fruit picking.

(3) Automatic navigation tests of the picking platform were conducted in the orchards. The results indicated that at 0.4 m/s, the maximum lateral deviation in straight-line path tracking was 101.5 mm and the maximum lateral deviation in U-shaped path tracking was 148.6 mm. The results demonstrated that the picking platform's path-tracking accuracy meets the requirements for orchard picking operations.

**Author Contributions:** Conceptualization, Q.Z. and S.H.; methodology, K.P. and S.W.; software, H.Y. and X.S.; validation, Y.Z.; writing—original draft preparation, S.H.; writing—review and editing, Q.Z.; visualization, K.P.; supervision, Q.Z. All authors have read and agreed to the published version of the manuscript.

**Funding:** This research was funded by the National Key R&D Program of China (2022YFD2001902).

**Institutional Review Board Statement:** Not applicable.

**Data Availability Statement:** The data presented in this study are available on request from the corresponding author.

**Conflicts of Interest:** The authors declare no conflict of interest.

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
