# Peer review of "Design and Test of an Automatic Navigation Fruit-Picking Platform"

_agriculture, doi:10.3390/agriculture13040882_

Round 1
Reviewer 1 Report
This manuscript is devoted to the problem of increasing the autonomy of the introduction of automation when operating a self-propelled platform when moving in the aisles of the garden. To improve the quality of the manuscript, I recommend making some adjustments.
1. It is desirable to supplement the review with the available autonomous platforms for robotic fruit harvesting, pay attention to the implementation of autonomous motion systems applicable to them.
2. It is desirable to improve the quality of the drawings and photographs.
3. To the diagram in Figure 10, it should be described for which limiting angles of lifting of the platform your calculation was made. And the diagram of Figure 10 looks like from a "textbook" when calculating the maximum angles of arrival, think about how you can improve it or remove it altogether.
4. It is unclear how your voice control system will react to background noise when fruit pickers communicate with each other, because when using such platforms, a large number of workers are supposed to be used.? Has the issue of reliability of the system functioning and the issue of false positives been considered? These data can be given in the manuscript.
5. The number of tests in the table is too small, in order to increase the correctness of the accuracy of the data obtained, it was more correct to give a larger number of tests.
6.It is necessary to give the dimensions of the aisles, the height of the trees, the dimensions of the crowns where the platform was tested?
7. Please describe in more detail the automatic driving navigation system between the rows of trees.
8. The output should be expanded by specifying more digital data received.
Reviewer 2 Report
Dear Authors, below details remarks to manuscript:
1) Abstract - specify the aim of work,
2) Introduction - (last part), as point above,
3) Figure 16. Schematic ... - explain the exact meaning of this figure (the content should be understandable to a non-specialist)
4) Chapter 7. Conclusions - in my opinion, these are not conclusions, but a summary and recommendations (discussion/summary of results); consider changing the chapter title to "Summary"
5) References - inconsistent with MDPI editorial standards
Reviewer 3 Report
This paper designs and tests of automatic navigation fruit picking platform with voice control, Beidou navigation and Stanley algorithm. Experimental results show the proposed system is correct and effective. There are some problems of theoretical and experimental analyses in this manuscript and it can be revised in the following aspects.
1. The whole paper should be carefully checked to avoid some possible typographical or grammatical errors. For example, in Abstract section, the positions of “Then” and “Secondly” should be exchanged; “equation” should be “inequation” above Equation (1); section 5.2 is missed.
2. In the experimental section, performance comparison with classical systems or methods are necessary to demonstrate the superiority of the proposed system.
